# A New Single-Pass SAR Interferometry Technique with a Single-Antenna for Terrain Height Measurements

**Min-Ho Ka [1,*], Pavel E. Shimkin [2], Aleksandr I. Baskakov [2] and Mikhail I. Babokin [3]**

[1] School of Integrated Technology, Yonsei Institute of Convergence Technology, Yonsei University, 21983 Seoul, Korea

[2] Radio-engineering Faculty, Moscow Power Engineering Institute, 111250 Moscow, Russia; shimkinpy@mpei.ru (P.E.S.); baskakovai@mpei.ru (A.I.B.)

[3] Joint Stock Company Aerocon, 140181 Zhukovsky, Russia; m.baboki57@mail.ru

* Correspondence: kaminho@yonsei.ac.kr; Tel.: +82-32-749-5840

**Abstract:** One of the prospective research topics in radar remote sensing technology is the methodology for designing an optimal radar system for high-precision two-dimensional and three-dimensional image acquisition of the Earth's surface with minimal hardware requirements. In this study, we propose a single-pass interferometric synthetic aperture radar (SAR) imaging technique with only a single antenna for the estimation of the terrain height. This technique enabled us to obtain terrain height information in one flight of the carrier, on which only one receiving antenna was mounted. This single-antenna single-pass interferometry required a squint angle look geometry and additional image synthesis processing. The limiting accuracy of the terrain height measurement was approximately 1.5 times lower than that of the conventional two-pass mode and required a longer baseline than two-pass interferometry to have an equivalent accuracy performance. This imaging method could overcome the temporal decorrelation problem of two-pass interferometry due to a short time gap in the radar echo acquisitions during two sub-aperture intervals. We compared the accuracy performance of the terrain height measurements of our method with the conventional two-pass interferometry. This comparison was carried out at various spectral bandwidths, degrees of surface roughness, and baseline lengths. We validated our idea with numerical simulations of a digital elevation map, and showed real extracted data of the terrain heights in the Astrakhan and Volga regions of the Russian Federation, obtained from airborne SAR with our single-antenna single-pass interferometry technique.

**Keywords:** remote sensing; synthetic aperture radar; interferometry; single-pass; single antenna; squint angle; limiting accuracy

---

## 1. Introduction

In recent years, the development of theoretical principles and technical capabilities in the construction of synthetic aperture radar (SAR) has allowed researchers to obtain radar images with resolutions in the order of meters or better. Further improvement of the geo-information content from airborne and spaceborne remote sensing systems requires further development of the theory and techniques for obtaining detailed terrain data and the formation of three-dimensional imagery. Currently, one of the main research topics is the analysis and design of prospective radar systems capable of providing high precision and high-resolution image synthesis, as well as three-dimensional imagery of the Earth's surface [1]. The need to measure the terrain at the same scale as that of the radar image arises because unknown terrain in the mapping area leads to an inaccurate determination of an

object's coordinates. The elevation (height) of objects or elements of the terrain leads to the distortion of radar images, such as the displacement of the object's position in the horizontal (azimuth) direction due to Doppler frequency shifts in the azimuth, and the formation of radar shadows (i.e., masking of individual objects due to screening by other objects). It is known [1–3] that relative terrain height measurements by airborne SAR, obtained while observing a selected area, require spatially separated receiving antennas mounted on the same platform or separate platforms. These antennas receive reflected signals illuminated by the transmitting front end (transmitter and antenna), and one can coherently process these two or more signals to determine the elevation information. This process adds a third dimension (height) to the two-dimensional radar image. Two typical methods to construct such a system are (1) two-pass or repeat-pass SAR interferometry with one receiving antenna mounted on board with a single or tandem platform [2,4–8]; and (2) single-pass interferometry with two or more receiving antennas onboard [9–12]. The single-pass method is sometimes called fixed-baseline InSAR and can obtain coherent echo signals simultaneously to overcome the temporal decorrelation problem, and can possibly obtain a real-time interferogram and three dimensional terrain map. However, single-pass SAR cannot avoid the limitation of a short baseline due to the size of the flying vehicle.

In this paper, we propose a single-pass interferometric synthetic aperture radar (SAR) imaging technique with only a single antenna, for the estimation of the terrain height. Single antenna single-pass interferometry requires squint angle look geometry and additional image synthesis processing. This technique can overcome several drawbacks of two-pass interferometry including temporal decorrelation, long revisit time, and non-optimal baseline geometry. At the same time, this mode requires only one receiving antenna and has no baseline length limitation, like in the fixed-baseline InSAR.

Single-antenna single-pass interferometry has been considered in several earlier publications [13–15]. In particular, Reference [13] presented the topography extraction of the surface of Venus by using interferometric processing of single-orbit Magellan SAR data. Recently, single-antenna single-pass interferometry has been tested using spaceborne and airborne SAR. Both spaceborne and airborne experimental tests have demonstrated the capability of this technique to measure the Earth's surface relief. The experimental data from spaceborne SAR was obtained from the S-band radar device Strizh, installed on the Kondor-E satellites launched in 2013 and 2014 [16]. The interferometry technique was implemented in spaceborne SAR as an experimental operational mode. Experimental data obtained using single-antenna single-pass airborne SAR is presented in References [16,17].

It must be mentioned that there is an alternative method for retrieving the height by using a single antenna squint-mode airborne SAR, which was proposed and experimentally verified in Reference [18]. The method proposed in these works was not interferometric, but based on Doppler frequency signal processing in squint SAR, where targets with different heights belong to different Doppler cones. The method from Reference [18] requires very precise SAR attitude determination and accurate Doppler centroid measurements.

In this study, we analyzed and compared the accuracy performance of two interferometric SAR imaging methods where a radar with one transceiver channel was installed on a single aircraft: (1) a single-antenna single-pass imaging mode with a linear flight trajectory, on which a synthetic interferometric baseline was formed; and (2) a two-pass imaging mode with close repeating passes, the distance between which forms the baseline of each synthetic aperture radar for the interferometry.

The rest of this paper is organized as follows. Sections 2 and 3 describe and derive geometrical and mathematical relations between the height of the terrain and interferometric phase difference for both imaging methods. Section 4 describes the error components in the terrain height measurement using interferometric SAR. In Section 5, the correlation coefficient is estimated between two complex datasets of synthesized signals in SAR images that generate interferograms. In Section 6, we calculate the limiting accuracy of interferometric measurement errors of the terrain height for each interferometry mode, and analyze the measurement performance as a function of several parameters of the radar system. Section 7 outlines the numerical simulation results of the single-antenna single-pass synthesized

Digital Elevation Map (DEM) model. We calculate optimal baseline conditions, synthesize the radar images, obtain interferograms, generate DEMs, and then calculate the measurement errors to show the validity of the obtained accuracy estimations. Section 8 briefly presents the experimental results from our previous work [17], demonstrating the validity of this idea with airborne SAR raw data for the Astrakhan and Volga regions in the Russian Federation. Finally, Section 9 summarizes this study.

## 2. Single-Antenna Single-Pass SAR Interferometry

In the proposed imaging geometry for single-antenna single-pass interferometry, spatial separation is achieved through the natural flight movement of the SAR carrier and by illuminating the antenna beam pattern on the area of interest of the Earth's surface. The two interferometric datasets are obtained in two consecutive observations, i.e., sub-apertures, of the area of interest under slightly different squint angles.

The principle of operation consists of the sequential observations of the surface at range $R_1$, azimuth angle $\alpha_1$, and look angle $\theta_1$ in the first session $L_1$, and then again at a different range $R_2$, different azimuth angle $\alpha_2$, and different look angle $\theta_2$, in the second session $L_2$. The radar carrier is assumed to move at a constant altitude $H$ and a constant speed $V$. The geometric parameters are shown in Figure 1, where $L_1$ and $L_2$ are the sizes of the sub-aperture lengths. The second sub-aperture observation session $L_2$ started when the carrier moves a distance equal to the baseline of the interferometer, $B$, which is defined by the slow time offset between the two sub-aperture observations.

We have to mention that the squint angles and look angles were very close even though it is exaggerated in the figure in order to explain the principle of the method, that is $\alpha_1 \approx \alpha_2$ and $\theta_1 \approx \theta_2$. The squint angle $\alpha_1$ generates a perpendicular component of the baseline and consequently enables us to obtain the height information of surface relief. The synthesis of the received signals in the receiving antenna during each sub-aperture, $L_1$ and $L_2$, is performed and the interferogram can be generated from the two sets of complex data for each sub-aperture synthesis.

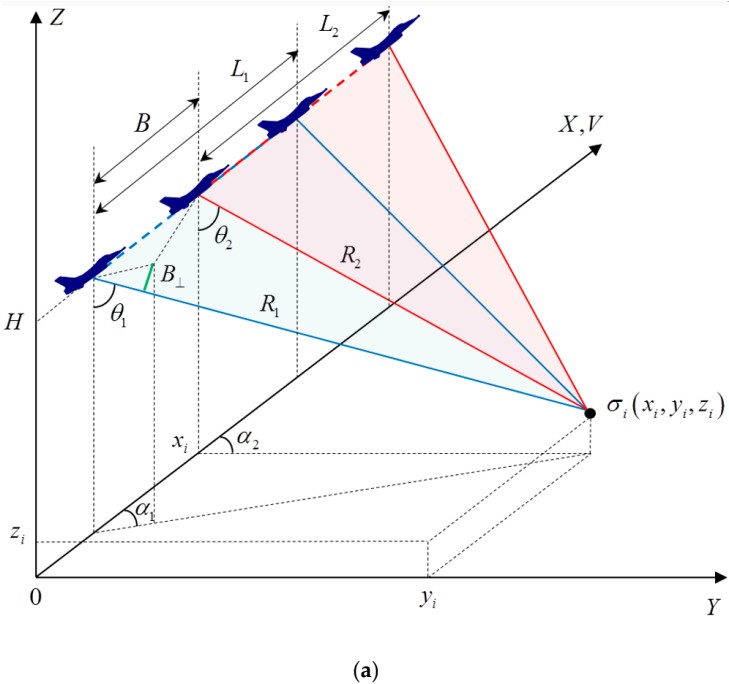

(a)

**Figure 1.** *Cont.*

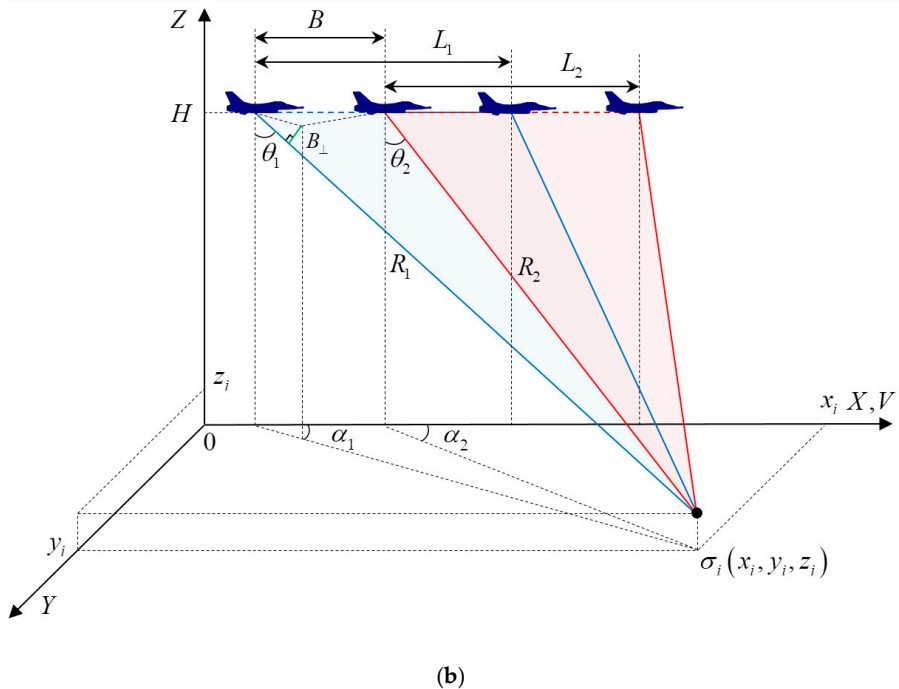

(**b**)

**Figure 1.** Imaging geometry of the single-antenna single-pass interferometric synthetic aperture radar (SAR): (**a**) 3-D from Z-Y view; (**b**) 3-D from Z-X view.

From the geometry in Figure 1, the following dependence of the height of the resolution element on the system parameters $z_i$, and the phase difference $\phi$, may be derived (see Appendix A):

$$z = H - R_1 \sqrt{1 - \left( \frac{R_1^2 + B^2 - (R_1 - \lambda\phi/(4\pi))^2}{2R_1 B \cos\alpha_1} \right)^2}.$$ (1)

If we assume that $B/R_1 \ll 1$ and $z \ll R_1$, approximate expressions for interferometric phase can be derived. The interferometric phase difference $\phi$ is approximately expressed as:

$$\phi \approx \frac{4\pi B}{\lambda} \sin\theta_1 \cos\alpha_1 + \frac{4\pi B_\perp}{\lambda R_1 \sin\theta_1} z,$$ (2)

where $B_\perp = B\cos\alpha_1 \cos\theta_1$ is the orthogonal baseline. The first term in Equation (2) corresponds to the systematic phase delay, which is similar to the phase shift between antenna array elements. This component is also called the "flat Earth phase contribution", i.e., the phase difference obtained under $z = 0$. The second term defines the sensitivity of the interferometric phase changes $\delta\phi$ to height changes $\delta z$:

$$\delta\phi \approx \frac{4\pi B_\perp}{\lambda R_1 \sin\theta_1} \delta z.$$ (3)

Due to the synthetic nature of the interferometric baseline, a single interval of synthesis may be used and split into subintervals (sub-apertures) at the digital processing stage to implement the interferometry. The advantages of this approach are as follows. First, the length of the baseline may be optimized for minimizing the height estimation error. Second, it becomes possible to flexibly resize the baseline of the interferometer in order to unwrap the phase using the multi-base method. In this case, Equations (1) and (2) remain unchanged.

## 3. Two-Pass Interferometry

Interferometric imaging on an area of interest can be carried out with the side-looking mode (for example, the strip map mode) by using a single radar on an aircraft in two repeat passes. It can be realized with the exact same radar systems on the same aircraft (or spacecraft) flying by in different passes.

In Figure 2, we assumed that there was a shift in the carrier trajectory. The altitude change in the second flight path is this $\Delta H$, and $\Delta Y$ is the shift in the direction transverse to the second flight path.

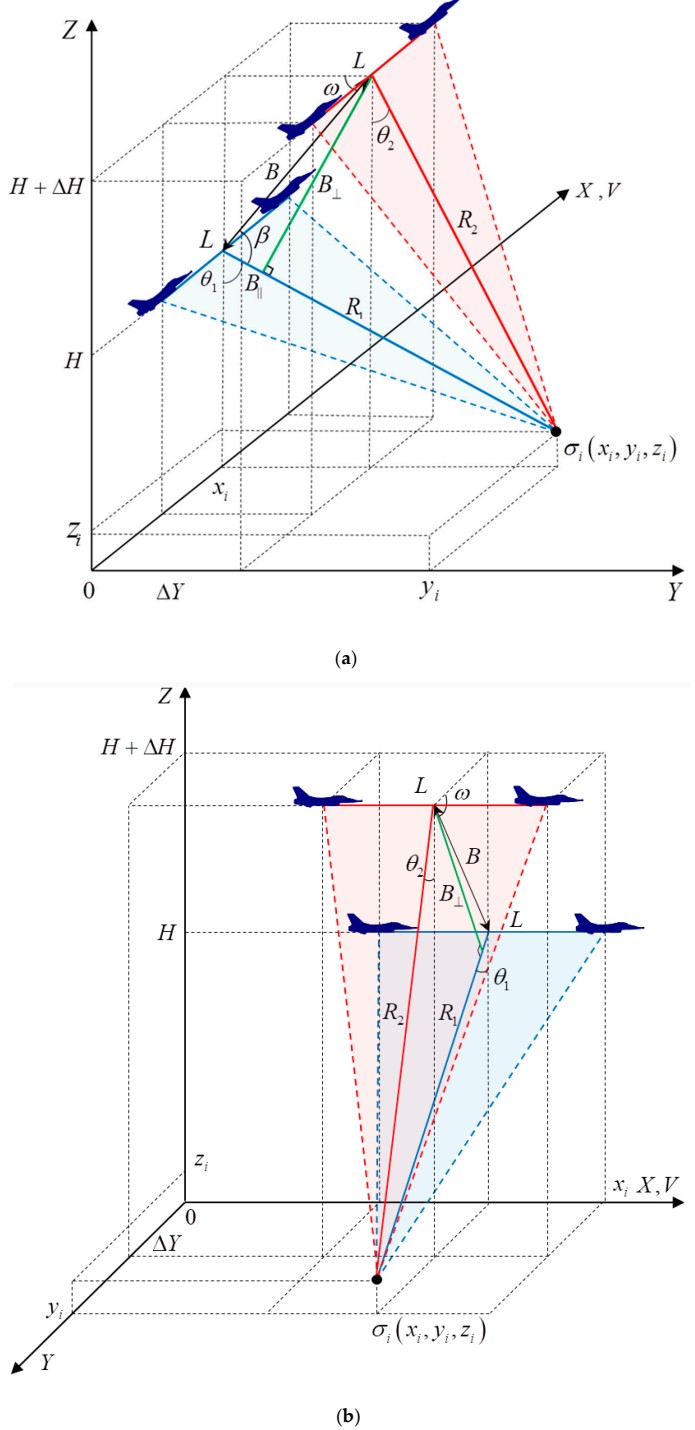

**Figure 2.** Imaging geometry of the two-pass interferometric SAR; (**a**) 3-D from Z-Y view; (**b**) 3-D from Z-X view.

From the geometry in Figure 2, the relationship between the height of the resolution element $z_i$, baseline $B$, height $H$, baseline tilt angle $\omega$ (the angle between the horizon and baseline $B$), measured distance $R_1$, and interferometric phase difference $\phi$ is as follows [2]:

$$z = H - R_1 \left( \cos\omega \cdot \sqrt{1 - \left( \frac{R_1^2 + B^2 - (R_1 - \lambda\phi/(4\pi))^2}{2BR_1} \right)^2} - \sin\omega \cdot \left( \frac{R_1^2 + B^2 - (R_1 - \lambda\phi/(4\pi))^2}{2BR_1} \right) \right). \quad (4)$$

## 4. Error Components in Terrain Height Measurements

Consider the case of the single-antenna single-pass or two-pass interferometric SAR. From Equations (1) and (4), it is obvious that the height of a resolution element $z_i$ is a function of the following parameters: altitude of the radar $H$, slant range $R_1$, squint angle $\alpha_1$, baseline tilt angle $\omega$, length of the baseline $B$, and interferometric phase difference $\phi$. Therefore, the overall variance of the height of the resolution element $\sigma_z^2$, characterizing the measurement error, can be written as follows, assuming no correlations between error components:

$$\sigma_z^2 = \sigma_{z\hat{\phi}}^2 + \sigma_{z\alpha_1}^2 + \sigma_{zH}^2 + \sigma_{zR_1}^2 + \sigma_{zB}^2, \quad (5)$$

for the single-pass interferometer, and:

$$\sigma_z^2 = \sigma_{z\hat{\phi}}^2 + \sigma_{z\omega}^2 + \sigma_{zH}^2 + \sigma_{zR_1}^2 + \sigma_{zB}^2, \quad (6)$$

for the two-pass interferometer, where $\sigma_{z\hat{\phi}}^2$, $\sigma_{z\alpha_1}^2$, $\sigma_{z\omega}^2$, $\sigma_{zH}^2$, $\sigma_{zR_1}^2$, and $\sigma_{zB}^2$ are the variance in the resolution element height due to the estimation error of the phase difference $\sigma_{\hat{\phi}}$, the measurement error of the squint angle $\sigma_{\alpha_1}$, the measurement error of the baseline tilt angle $\sigma_{\omega}$, the measurement error of the altitude $\sigma_H$, the measurement error of the slant range $\sigma_{R_1}$, and the measurement error of the baseline length $\sigma_B$, respectively.

Determination of the most efficient imaging mode for the interferometric SAR observation was conducted to find the imaging parameters that minimized the errors listed above. Since the limiting accuracy of the measurement was defined by $\sigma_{\hat{\phi}}$ and the other components of the measurement introduced systematic errors in $\sigma_z^2$, only the analysis of the error associated with the estimation of interferometric phase difference $\hat{\phi}$ was considered in this study.

For the two-pass interferometer, $\sigma_{z\hat{\phi}}$ is known from Reference [2], and in our notations, it is expressed as follows:

$$\sigma_{z\hat{\phi}}^{two} = \frac{\lambda H \tan\theta_1}{4\pi B \cos(\theta_1 - \omega)} \sigma_{\hat{\phi}}. \quad (7)$$

For the single-antenna single-pass interferometry, differentiation of Equation (1) results in the following expression for $\sigma_{z\hat{\phi}}$:

$$\sigma_{z\hat{\phi}}^{single} = \frac{\lambda H \tan\theta_1}{4\pi B \cos\alpha_1 \cos\theta_1} \sigma_{\hat{\phi}}. \quad (8)$$

Equations (7) and (8) can be combined into a single equation:

$$\sigma_{z\hat{\phi}} = \frac{\lambda H \tan\theta_1}{4\pi B_\perp} \sigma_{\hat{\phi}}, \quad (9)$$

where $B_\perp$ is the perpendicular projection of the baseline (the effective baseline):

$$B_\perp = B \cos\alpha_1 \cos\theta_1 \quad (10)$$

for the single-antenna single-pass mode, and

$$B_\perp = B\cos(\theta_1 - \omega) \tag{11}$$

for the two-pass mode.

According to References [2,4], the interferometric phase difference can be calculated using maximum likelihood estimation and the standard deviation of the interferometric phase difference $\sigma_{\hat\phi}$ can then be evaluated using the following expression:

$$\sigma_{\hat\phi} = \frac{1}{\sqrt{2N}}\frac{\sqrt{1-\gamma^2}}{\gamma}, \tag{12}$$

where $N$ is the number of incoherent integration and $\gamma$ is the correlation coefficient for the two received signals in the interferometer.

## 5. Estimation of the Correlation Coefficient

The interferometric coherence or correlation coefficient between the two complex datasets of synthesized signals in SAR that generate an interferometric pair, in general, consists of several decorrelation factors [4,19].

The first factor is the decorrelation due to difference of incidence angles. This factor is one of the spatial decorrelations, and for a regular surface it is expressed as [19]:

$$\gamma_{spatial,reg} = 1 - \frac{2B_\perp}{\lambda R \tan\theta_1}\Delta r, \tag{13}$$

where $\Delta r$ is the slant range resolution. Substituting the corresponding expressions—Equations (10) and (11)—for $B_\perp$ into Equation (13), we obtain for single-antenna single-pass interferometry:

$$\gamma_{spatial,reg}^{single} = 1 - \frac{2B\cos\alpha_1\cos\theta_1}{\lambda R \tan\theta_1}\Delta r, \tag{14}$$

and for two-pass interferometry:

$$\gamma_{spatial,reg}^{two} = 1 - \frac{2B\cos(\theta_1 - \omega)}{\lambda R \tan\theta_1}\Delta r. \tag{15}$$

Accounting for surface or volume irregularities requires an appropriate model for the scatterers. For this study, we modeled the surface roughness through a large number of random independent scattering elements. We assumed the surface to be composed of distributed radar targets and modeled a set of independent partial reflectors inside the resolution element. We also assumed a normal density distribution for the height of the partial reflectors inside the resolution element. The spatial decorrelation factor accounting for surface roughness and assuming such a model has been estimated as [10,11]:

$$\gamma_{spatial} = \gamma_{spatial,reg}\cdot\gamma_{surface}, \tag{16}$$

where:

$$\gamma_{surface} = \exp\left[-2\pi^2\left(\frac{\sigma_h B_\perp}{\lambda R \sin\theta_1}\right)^2\right], \tag{17}$$

and $\sigma_h$ is the root mean square ordinate of the small-scale roughness of the Earth's surface. Substituting correspondent Equations (10) and (11) for $B_\perp$ into Equation (17), we obtain the following for the single-antenna single-pass interferometer:

$$\gamma_{surface}^{single} = \exp\left[-2\pi^2\left(\frac{\sigma_h B \cos\alpha_1 \cos\theta_1}{\lambda R \sin\theta_1}\right)^2\right],$$
(18)

and for the two-pass interferometer:

$$\gamma_{surface}^{two} = \exp\left[-2\pi^2\left(\frac{\sigma_h B \cos(\theta_1 - \omega)}{\lambda R \sin\theta_1}\right)^2\right].$$
(19)

It should be mentioned that the considered scattering model of the surface does not adequately describe vegetated and snowy regions. For such regions, other models may be used that represent a large number of homogeneous independent scatterers distributed in a layer over the ground surface [20]. The coherence loss due to the volume scattering in the aforementioned layer is known as volume decorrelation [21]. In this study, we only took surface irregularities into account. Considering a more complex model, including scatterers distributed over volume, is the subject of further work.

Another factor is temporal decorrelation, which is due to changes on the surface between the two observations ($\gamma_{temporal}$). For this study, we did not consider the temporal decorrelation factor in order to analyze the measurement performance of the radar systems, and assumed $\gamma_{temporal} = 1$ for both cases. This temporal decorrelation factor still gives useful information about surface change phenomena, for example, change detection [7,8,22], and is widely used in various applications of radar remote sensing.

The next factor is decorrelation due to thermal noise in the system ($\gamma_{thermal}$). The effect of thermal noise reduces the coherence in both cases by a factor:

$$\gamma_{thermal} = \frac{1}{1 + SNR^{-1}},$$
(20)

where *SNR* is the signal-to-noise ratio.

In the case of single-antenna single-pass interferometry, there is another important factor, which decreases the correlation related to the deployment of the interferometry baseline along the flight path of the carrier. It is obvious that the surface is illuminated by electromagnetic waves from two different azimuth angles $\alpha_1$ and $\alpha_2$ in a single-pass case. This effect decreases the correlation of the image pair caused by the rotation of each resolution element through a certain angle, which directly depends on the length of the baseline and squint angle $\alpha_1$ and is defined as $\gamma_{rotation}$. The decorrelation due to rotation has been estimated as [4]

$$\gamma_{rotation} = 1 - \frac{2\sin\theta_1 \Delta x}{\lambda} d\psi,$$
(21)

where $d\psi$ is the difference of azimuth angles of the two receivers and $\Delta x$ is the azimuth resolution. From the geometry in Figure 1, the rotation angle can be expressed as

$$d\psi = \arctan\left(\frac{B \sin\alpha_1}{H \tan\theta_1 - B \cos\alpha_1}\right).$$
(22)

Therefore, in a single-pass interferometer,

$$\gamma_{rotation}^{single} = 1 - \frac{2\Delta x \sin\theta_1}{\lambda} \cdot \left|\arctan\left(\frac{B \sin\alpha_1}{H \tan\theta_1 - B \cos\alpha_1}\right)\right|.$$
(23)

In this work, we considered the simple case for two-pass interferometry when the baseline has no projection on the *X* axis in Figure 2. The azimuth angle for both receivers is thus the same. Therefore, for the two-pass interferometer $\gamma_{rotation}^{two} = 1$.

The correlation coefficient including the aforementioned effects represents a product of all of the decorrelation factors:

$$\gamma = \gamma_{spatial} \cdot \gamma_{thermal} \cdot \gamma_{rotation}. \tag{24}$$

It should be noted that the aforementioned effects do not cover all of the factors that affect the correlation between the interferometric datasets. There are numerous additional sources of decorrelation. The coherence may be degraded due to quantization effects [23], processing errors [24], and other systematic error sources. In this work, we focused on the effects that resulted in different decorrelation factors due to the different geometries considered in single-antenna single-pass and two-pass interferometry; namely, $\gamma_{spatial}$ and $\gamma_{rotation}$.

## 6. Limiting Accuracy of the Interferometric Measurements

After calculating the correlation coefficients (Equation (24)) as a combination of the decorrelation factors, and substituting them into Equation (12), we obtained the expression for the measurement error, the root mean square (RMS) phase difference. By substituting Equation (12) into Equation (9), we calculated the measurement error of the terrain height for each interferometry mode.

It is necessary to set specific numerical values to analyze the dependence of the RMS error of the terrain height estimation on the length of the interferometric baseline. We considered the dependence of the limiting accuracy $\sigma_{z\hat{\phi}}$ for the two modes of interferometric SAR as centimeter wavelengths, which is widely used in radar remote sensing. We assumed that the aircraft carrier was flying at an altitude $H = 5$ km and velocity $V = 250$ m/s; the carrier frequency was $f_c = 10$ GHz; the bandwidth of the sensing signal $\Delta f$ ranged from 150 MHz to 300 MHz; the corresponding slant range resolution, determined from $\Delta r = c/(2\Delta f)$, ranged from 0.5 to 1 m; the azimuth resolution $\Delta x$ was 0.5 m; the look angle $\theta_1$ was 45°; the squint angle $\alpha_1$ was 30° ($\alpha_2 = 30.05°$); the baseline tilt angle $\omega$ was 45°; the signal-to-noise ratio *SNR* was 10 dB; and the number of looks *N* was 4.

We compared the results with various spectral bandwidths of the signals and different degrees of small-scale surface roughness. For single-antenna single-pass and two-pass interferometry, the degree of fluctuations of the small-scale surface roughness $\sigma_h$ within the resolution element, for a relatively short baseline, have little effect on the RMS error of the height estimation, but the influence of surface roughness is evident for a relatively long baseline (Figure 3a).

As we increase the RMS roughness of the surface further, the estimation error of the height measurement increases more rapidly for the two-pass interferometry mode when compared to the single-pass mode, as shown in Figure 3b. Thus, single-antenna single-pass interferometry is more stable against rapid changes in the RMS roughness of the surface.

As shown in Figure 4a, when we increase the bandwidth of the sensing signal, the length of the optimal baseline for interferometry is significantly increased, and for the single-antenna single-pass interferometric SAR, this range is wider than that for the two-pass mode. Increasing the bandwidth of the sensing signal leads to a decrease in the height estimation error in both cases, but this increase is less effective for single-antenna single-pass interferometry than for two-pass interferometry, as shown in Figure 4b.

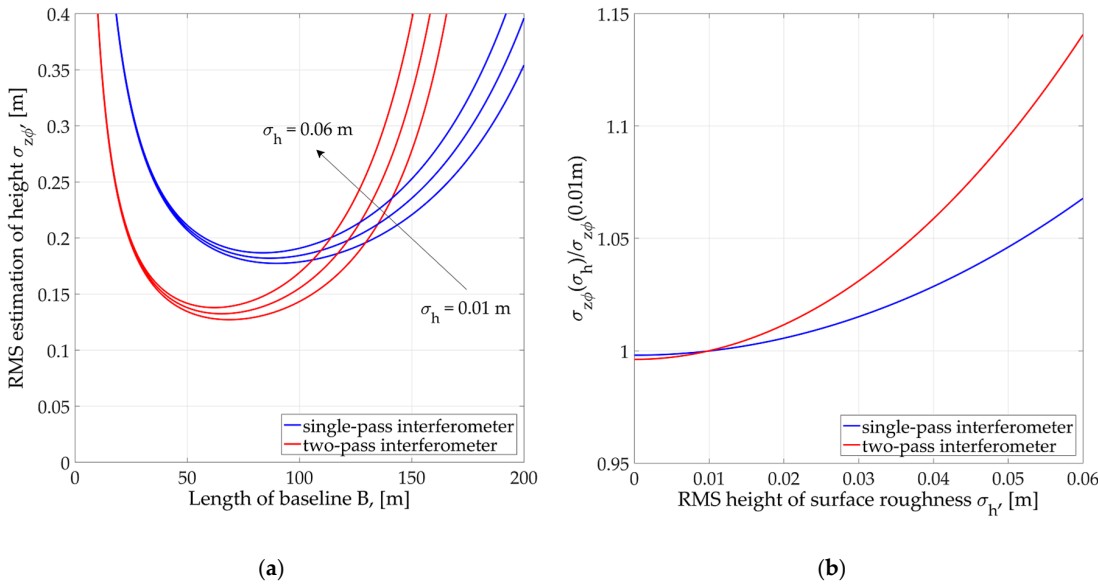

**Figure 3.** (**a**) Comparison of the height estimation error on the baseline length for the two interferometry modes at different values of the root mean square (RMS) ordinate for small-scale roughness $\sigma_h$. (**b**) Ratio of height estimation error as a function of the RMS heights for small-scale roughness $\sigma_h$ relative to $\sigma_h = 0.01$ m, under the same parameter values.

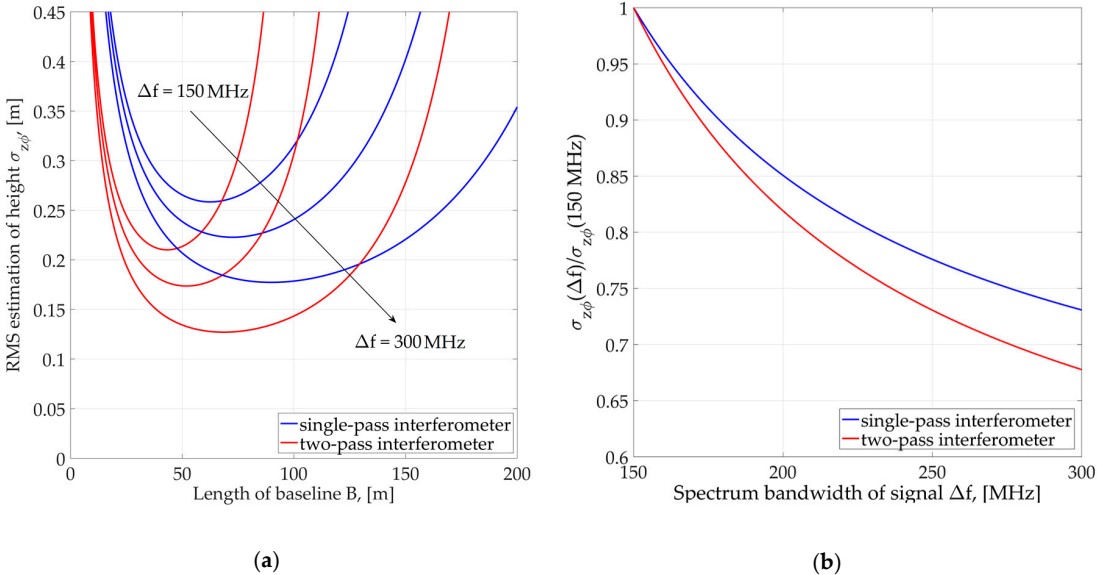

**Figure 4.** (**a**) Comparison of the height estimation error with respect to the baseline for the two interferometry techniques, with different bandwidths of the signal $\Delta f$. (**b**) Ratio of the height estimation error as a function of the bandwidth of the signal $\Delta f$, relative to $\Delta f = 150$ MHz, with the same parameter values.

By substituting the required operating frequency of the SAR transmitter and the azimuth and range resolution of existing SAR systems (Table 1) of the aircrafts and unmanned vehicles, leaving other parameters unchanged, we can obtain the values of the limiting accuracy of the measurement of the single-antenna single-pass interferometry with squint angle.

**Table 1.** Airborne/UAV-borne spotlight mode SAR platforms with achievable measurement accuracy.

| Sensor | Agency/Country | Band | Azimuth/Range Resolution, m | Limiting Accuracy, m |
|---|---|---|---|---|
| Lynx | Sandia/USA | Ku | 0.1/0.1 | 0.09 |
| GH | Northrop/USA | X | 1.8/1.8 | 0.55 |
| MiniSAR | Sandia/USA | Ka/Ku/X | 0.1/1 | 0.22 |
| LiMIT | MIT/USA | X | <1/<1 | <0.31 |
| CP-140 | LM/Canada | X | <1/<1 | <0.31 |
| I-MASTER | Tales-Astrium/UK | Ku | <1/<1 | <0.31 |
| Mini-SAR | TNO-FEL/Netherland | X | 0.05/0.05 | 0.07 |
| PAMIR | FHR-FGAN/Germany | X | 0.1/0.1 | 0.08 |

Table 1 lists the SAR with the spotlight mode necessary for the operation of the proposed interferometry imaging scheme. Table 2 summarizes the SAR systems that can potentially be modified to operate in the spotlight mode. The calculated values of the limiting accuracy of the measurements with the corresponding parameters of the SAR are summarized in the right columns of Tables 1 and 2 and are shown in Figure 5.

**Table 2.** Airborne/UAV-borne SAR platforms with possible modification to spotlight mode and their achievable measurement accuracy.

| Sensor | Agency/Country | Band | Azimuth/Range Resolution, m | Limiting Accuracy, m |
|---|---|---|---|---|
| C/X-SAR | CCRS/Canada | X/C | 0.9/6 | 1.24 |
| AIRSAR | NASA/USA | C/X/L | 0.6/3 | 0.64 |
| E-SAR | DLR/Germany | X/C/S/L/P | 0.3/1 | 0.24 |
| F-SAR | DLR/Germany | X/C/S/L/P | 0.3/0.2 | 0.12 |
| Pi-SAR | NICT, JAXA/Japan | X/L | 0.37/3 | 0.62 |
| EMISAR | DCRS/Denmark | C/L | 2/2 | 0.61 |
| PHARUS | TNO-FEL/Netherland | C | 1/3 | 0.68 |
| Ingara | DSTO/Australia | X | 0.15/0.3 | 0.12 |
| RAMSES | ONERA/France | Ka/Ku/X/C/S/L/P | 0.12/0.12 | 0.09 |
| DBSAR | NASA/USA | L | 10/10 | 3 |
| UAVSAR | NASA/USA | L | 1/1.8 | 0.46 |

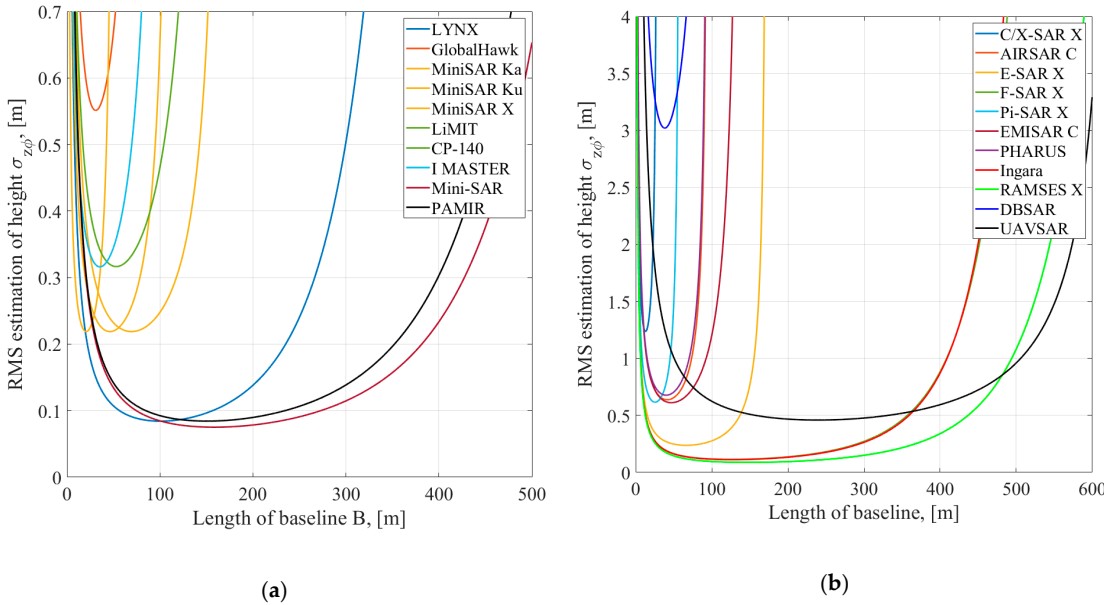

(**a**)                      (**b**)

**Figure 5.** (**a**) RMS estimation of height for airborne/UAV-borne from Table 1; and (**b**) RMS estimation of height for airborne/UAV-borne from Table 2. ($\sigma_h = \lambda$).

## 7. Numerical Simulations

Numerical simulations were carried out to demonstrate the accuracy of estimation using single-antenna single-pass interferometry.

### 7.1. Structure of the Simulation Model

The software package MATLAB was used as the simulation platform and the primary toolset for the radar simulation in this software was the Phased Array System Toolbox. The simulation process can be divided into the following stages:

1. Setting the DEM and its parameters;
2. Setting the parameters of the interferometric SAR system;
3. Generation of the echo signals along the path, their processing, and the synthesis of the radar images corresponding to the two simulation cases;
4. Calculation of the interferometric phase difference (IPD); and
5. Interferometric processing to obtain an elevation map as the final output.

If the interferometric baseline is smaller than the synthetic aperture length, only one observation interval should be used, and the interval should be split into sub-apertures during the digital signal processing stage [17]. In the following subsections, we overview each of these steps.

### 7.2. Digital Elevation Model

At this stage, the terrain features were generated according to the phenomenological surface model [25]. Each resolution element $\Delta x \times \Delta y$ on the Earth's surface was represented by a set of normally distributed partial scatterers, on which scattering conditions known from the experimental results were imposed.

For the simulation of single-antenna single-pass interferometry, we used the DEM shown in Figure 6a, where the center of the observed terrain had the coordinates $X_C = 4330$ m and $Y_C = 2500$ m. The optical image of the DEM is shown in Figure 6b.

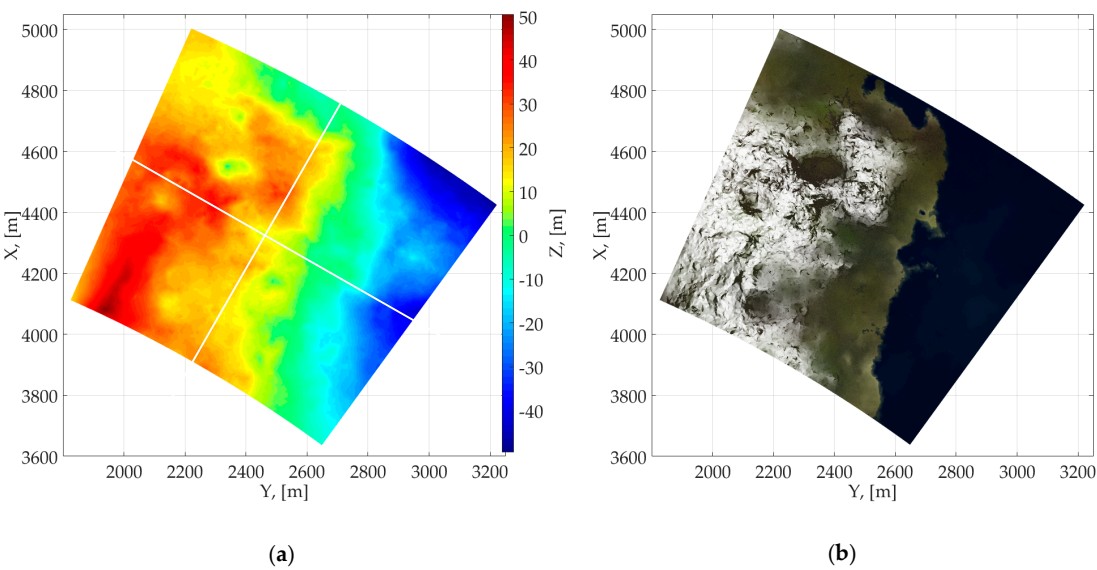

(**a**)　　　　　　　　　　　　　　　　　　　　　　　(**b**)

**Figure 6.** (**a**) DEM of the surface for the single-antenna single-pass interferometer; and (**b**) Optical image of the modeling surface.

We assumed a phased antenna array with azimuth and elevation beam widths of 7.5° and 5.5°, respectively. Taking the distance to the center of the observation area $R_C \approx 7.5$ km, we obtained an observation area of $1 \times 1$ km². The surface that we modeled represents a gently sloping gullied and

hilly structure with dry soil, which has the RMS ordinate of the small-scale roughness $\sigma_h = 2$ cm and an elevation level ranging from $-60$ m to 80 m.

We used the radar cross-section (RCS) model $\sigma^0$ for various surface types. This model considers the RMS ordinate of the small-scale roughness $\sigma_h$, the angle of incidence on the surface $\theta$, and wavelength $\lambda$. The RCS has the following form [26]:

$$\sigma^0(\theta, \sigma_h, \lambda) = A\left(\frac{\pi}{2} - \theta + C\right)^B \exp\left[-D\Big/\left(1 + \frac{0.1\sigma_h}{\lambda}\right)\right], \tag{25}$$

where $A, B, C,$ and $D$ are parameters of the empirical model. The numerical values of these parameters for some types of the surface are given in Reference [26] for the frequency range from 3 GHz to 95 GHz. Figure 7 shows the dependence of the specific RCS of some types of surface at 10 GHz frequency and 2 cm RMS ordinate of the small-scale roughness.

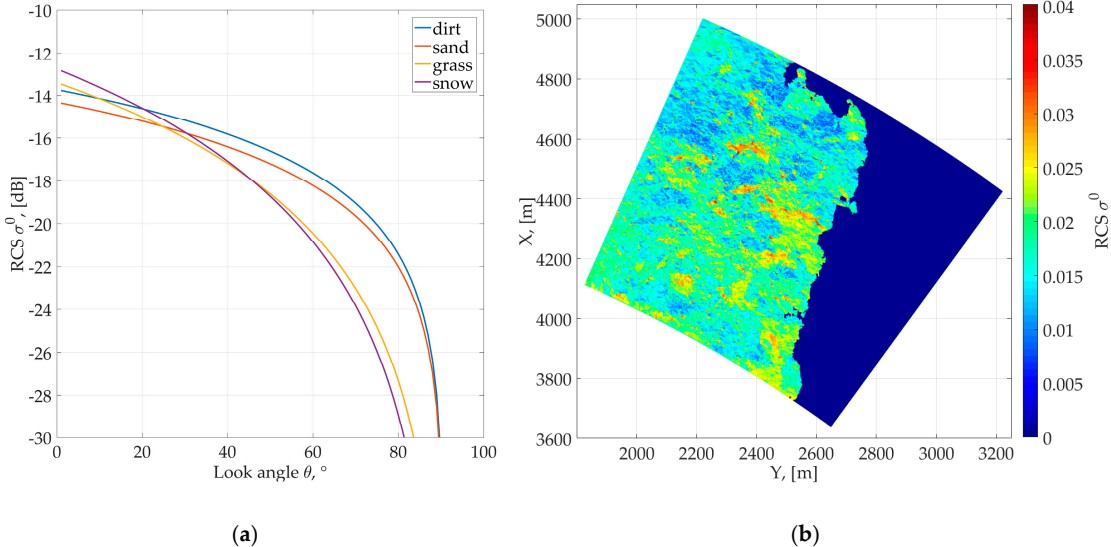

(a)　　　　　　　　　　　　　　　　　　　　　　　　　　　　　(b)

**Figure 7.** (**a**) RCS of the modeling surface for some types of surfaces. (**b**) RCS for the modeled area, using the DEM and the appropriate surface types.

### 7.3. Parameters of Interferometric SAR

For these simulations, we used the parameters of an airborne system, with an operational wavelength of $\lambda = 3$ cm. We used a linear-frequency modulated (LFM) signal with a bandwidth of $\Delta f = 30$ MHz, which gives a slant range resolution of $\Delta r = 5$ m, and, correspondingly, an azimuth resolution of $\Delta y = \Delta r / \sin \theta_1 = 7$ m. To achieve the azimuth resolution $\Delta x = \Delta y = 7$ m, a synthetic aperture $L = \lambda R_C / (2\Delta x \sin \alpha_1) \approx 46$ m was used. The pulse duration (or length) may be defined by choosing an off-duty factor $Q = \text{PRI}/\tau$, where PRI is the pulse repetition interval. Assuming that $Q = 10$, then, if $\text{PRI} = 2R_{\max}/c \approx 50$ µs, we find a pulse length of $\tau = 5$ µs. The pulse repetition interval of the sensing signal is calculated from the unambiguity criteria for both of the azimuth ranges:

$$\frac{2R_{\max}}{c} + \tau + t_a \leq \text{PRI} \leq \frac{d_a}{2V}, \tag{26}$$

where the velocity of the carrier is $V = 250$ m/s, the antenna size in the azimuth plane is $d_a = 1$ m, and the antenna duplexer switching time is $t_a = 3$ µs. Therefore, we obtained the requirement 58 µs $\leq$ PRI $\leq$ 2000 µs. Thus, we further assumed that PRI = 60 µs.

With the parameters of the interferometric SAR calculated as above, we obtained the following dependence of the limiting accuracy for height estimation of the terrain surface, on the interferometric baseline length. For the same parameters, there was a limiting accuracy of approximately 2 m and

the optimal baseline length *B* was 7.8 m. As previously mentioned, under such conditions ($B < L$), a single synthesis interval was used for synthesizing radar image pairs.

### 7.4. Synthesis of Radar Images and Interferometric Processing

The modeling, processing, and synthesis of the trajectory signal were carried out with the back-projection algorithm [27]. The outputs of this stage were pairs of Single-Look Complex (SLC) images, SLC-1 and SLC-2, for the studied DEM. The magnitude and the phase difference of one of the SAR images are shown in Figure 8.

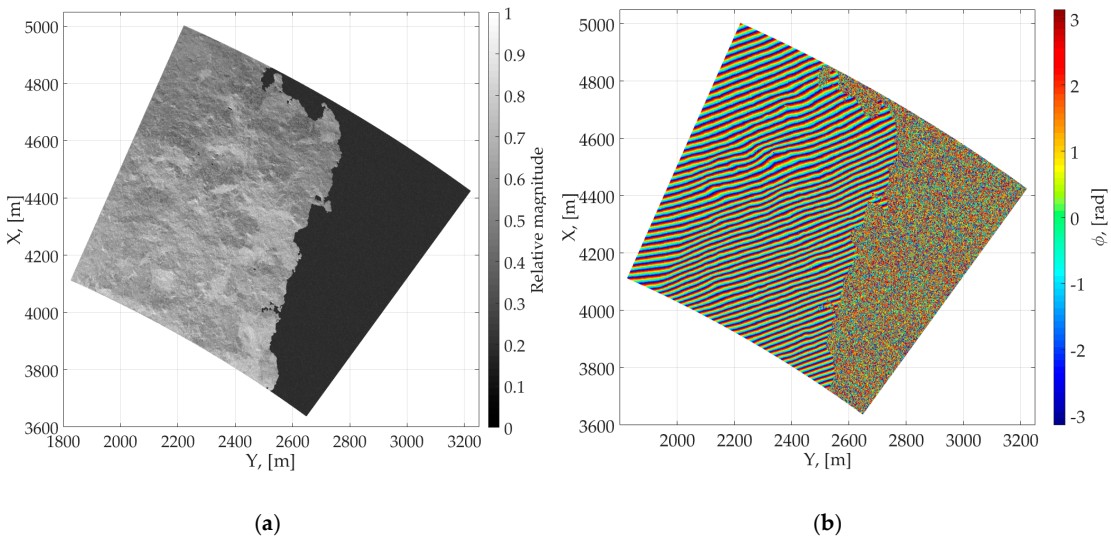

(a) (b)

**Figure 8.** (**a**) Magnitude of SLC image SLC-1 for the single-antenna single-pass interferometer. (**b**) Interferometric phase difference (IPD) for the single-antenna single-pass interferometer.

If we denote the radar images obtained during the two intervals or sub-apertures of observation as $P_1$ and $P_2$, we can then obtain an interferogram from their pixel-by-pixel complex conjugate multiplication:

$$I_{P_1 P_2}(x, y) = P_1(x, y) P_2^*(x, y) = \left| P_1(x, y) \right| \cdot \left| P_2(x, y) \right| \exp\left\{ j \left[ \phi_{P_1}(x, y) - \phi_{P_2}(x, y) \right] \right\}. \tag{27}$$

The interferometric phase difference can be defined as the argument of the multiplication result:

$$\phi_{P_1 P_2}(x, y) = \arg\left\{ \sum_{n=1}^{N} I_{P_1 P_2}(x, y) \right\}, \tag{28}$$

or by the arctangent function:

$$\phi_{P_1 P_2}(x, y) = \arctan\left\{ \mathrm{Im}\left[ \sum_{n=1}^{N} I_{P_1 P_2}(x, y) \right] \Big/ \mathrm{Re}\left[ \sum_{n=1}^{N} I_{P_1 P_2}(x, y) \right] \right\}. \tag{29}$$

Figure 8b illustrates the interferometric phase difference (IPD) for the simulated DEM.

The standard interferometric processing followed that described in References [3,19,28] and included:

1. Elimination of the linear phase component along the range by subtracting the phase of the flat Earth from the IPD of the DEM; removing to the effects of the flat surface of the Earth (Figure 9a);

2.　Elimination of the phase ambiguity. Since the IPD may significantly exceed two during elevation changes, the recovery of the true phase difference from the IPD reduces to the interval $(-\pi, \pi]$, and must be processes in an approach known as phase unwrapping [28] (Figure 9b); and

3.　Scaling of the unwrapped IPD and generation of the DEM according to the unambiguous relationship between terrain elevation and change of IPD.

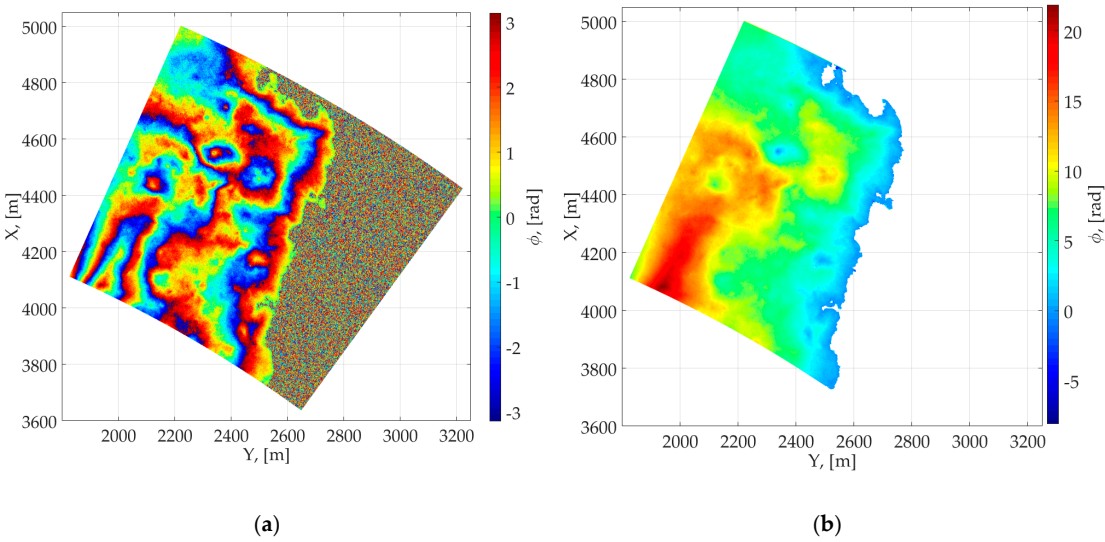

(**a**)　　　　　　　　　　　　　　　　　　　　　　　　　　(**b**)

**Figure 9.** Evolution of the IPD for the single-antenna single-pass interferometer during processing and analysis. (**a**) IPD after removing the flat Earth IPD; and (**b**) unwrapped IPD.

We assessed the measurement errors of single-pass and two-pass in two sections from the middle of the frame of Figure 6a. We calculated standard deviation of each cut to compare error of these two methods. The RMS error of the phase was 0.44 rad and the terrain RMS error of height with respect to the DEM model for the observation area was 2.05 m and 2.1 m for single-antenna single pass case. The accuracy is 1.36 and 1.49 times (about 1.5 times) larger than two-pass interferometry (Figure 10).

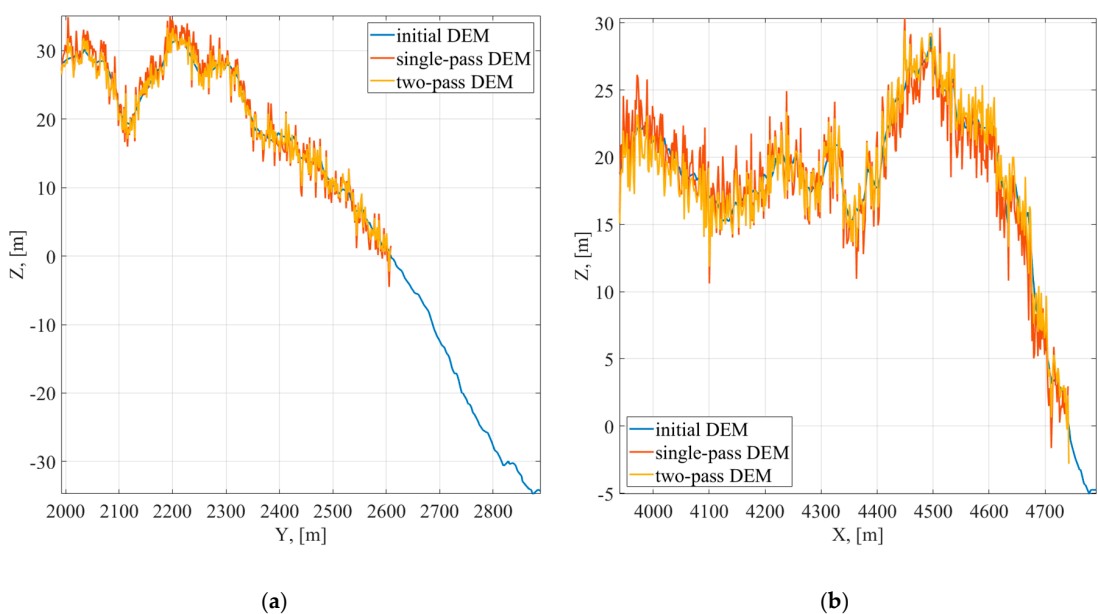

(**a**)　　　　　　　　　　　　　　　　　　　　　　　　　　(**b**)

**Figure 10.** (**a**) Compare DEM from Y cut; $\hat{\sigma}_z^{single-pass} \approx 2.05$ m, $\hat{\sigma}_z^{two-pass} \approx 1.51$ m and (**b**) Compare DEM from X cut; $\hat{\sigma}_z^{single-pass} \approx 2.1$ m, $\hat{\sigma}_z^{two-pass} \approx 1.43$ m.

## 8. Experiments Using Single-Antenna Single-Pass Interferometry with Airborne SAR

In order to demonstrate the validity of our idea, we present the images obtained from the experiments in our previous work [17] and briefly describe the experimental conditions. In this experiment, we extracted the terrain height data from two sets of interferograms using the single-antenna single-pass interferometric SAR method with high squint angles of 40° and 30°.

During this experiment, an airborne SAR recorded the data of two regions: (1) the Big Bogdo Mountain in the Astrakhan region of the Russian Federation (height of the mountain is 149.6 m); and (2) the Volga Hydroelectric Power Station (HPS) in the Russian Federation (height of the dam is 44.5 m). In this experiment, only a single synthetic aperture interval was used. This interval was divided into two sub-intervals (sub-apertures) during the signal processing stage. The experimental parameters corresponding to each location are presented in Table 3.

**Table 3.** Experimental parameters and locations.

| Parameters | Big Bogdo | Volga HPS |
|---|---|---|
| Distance to the image center, km | 60 | 40 |
| Aircraft flight height, km | 5 | 8 |
| Aircraft velocity, m/s | 150 | 200 |
| Squint angle, deg | 40 | 30 |
| Wavelength, cm | 3 | 3 |
| Pulse repetition interval, ms | 1 | 1 |
| Synthetic aperture length, m | 255 | 340 |
| Range resolution, m | 15.1 | 15.3 |
| Cross-range resolution, m | 5 | 3.2 |
| Interferometric baseline, m | 23.7 | 32 |
| Limiting accuracy, m | 8.7 | 1.6 |

The images obtained in this experiment are shown in Figures 11 and 12 along with the corresponding optical images from Google Earth. The left figures present the magnitude SAR images and the right figures show the elevation maps obtained from the data with the single-antenna single-pass interferometric SAR processing and a 40° squint angle. The relative positions of the mountain peak and the dam in the image are labeled on the elevation maps. The Z-coordinate on the labels corresponds to the estimated heights. As we can see from the figures, the obtained heights of the mountain peak and the dam were close to the real values.

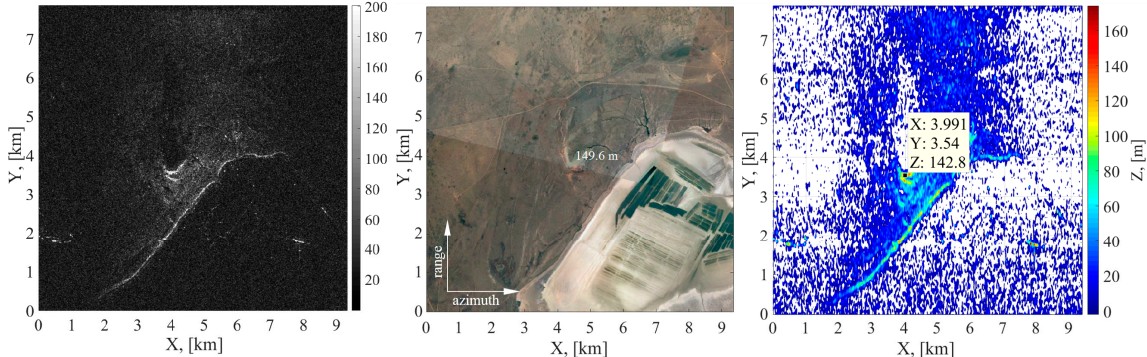

**Figure 11.** Single antenna single-pass SAR images of Big Bogdo Mountain.

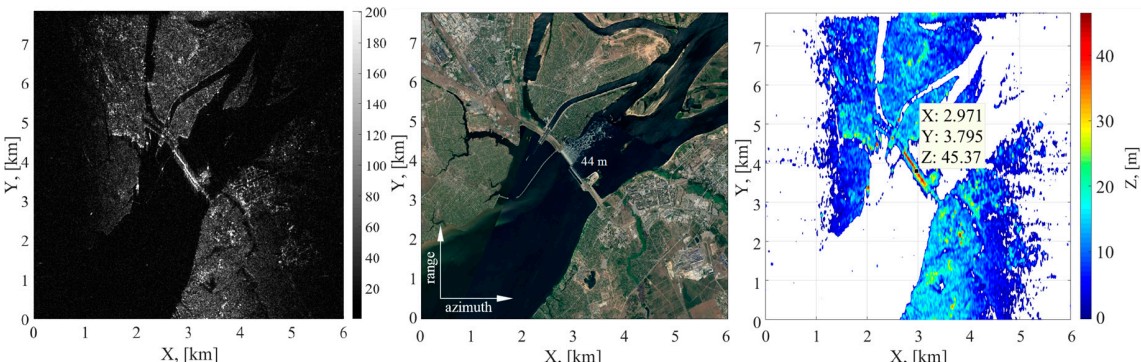

**Figure 12.** Single-antenna single-pass SAR images of the Volga Hydroelectric Power Station (HPS).

## 9. Conclusions

Single-antenna single-pass interferometry requires an additional algorithm for synthetic aperture signal processing. In this paper, we showed the mathematical relations between the height of the terrain and IPD for both single-pass and two-pass imaging geometries, in Sections 2 and 3, respectively. We then presented the experimental results from our previous work of high squint angle single-pass interferometric SAR processing with airborne radar raw data for two regions: Big Bogdo Mountain in the Astrakhan region and the Volga Hydroelectric Power Station, both in the Russian Federation. We showed that the height measurement values were close to the real values.

A numerical simulation was conducted to evaluate the accuracy performance of the single-antenna single-pass imaging mode using a terrain DEM model. We carried out interferometric signal processing, including the interferometric phase difference calculation, phase unwrapping, DEM generation, RMS error comparison of the phase, and DEM height estimation. The results demonstrated that this new method is a promising approach for operational interferometric SAR observations of terrain surfaces for 3D mapping, DEM generation, and other uses.

The limiting accuracy of the single-antenna single-pass interferometry in the height estimation of a local terrain was about 1.5 times lower than that of the conventional two-pass mode. Additionally, in order to minimize the measurement error for single-antenna single-pass interferometry, it is necessary to have a longer baseline compared to the two-pass mode, with all of the other parameters being equal. The additional requirement of single-antenna single-pass interferometry is the need of a special algorithm for squint angle synthetic aperture signal processing.

Despite the limitations and technical challenges, single-antenna single-pass SAR interferometry is a promising approach for various radar interferometry applications, such as the three-dimensional terrain mapping as shown in this study. The temporal decorrelation problem of the two-pass SAR interferometry constraint, due to the considerable time gap, can be overcome. This can be a unique and effective method to obtain real-time remote sensing information from one flying vehicle with minimal hardware, as it only requires a single transmitter and receiver channel structure, which is particularly important for small and lightweight flying vehicles such as drones.

**Author Contributions:** M.-H.K. and A.I.B. performed the mathematical derivations, edited and revised the manuscript. P.E.S. performed the simulation, analysis of the experiments, and edited the manuscript. M.I.B. performed the analysis of the experiments.

**Funding:** This research was supported by the MSIP (Ministry of Science, ICT, and Future Planning), under the "ICT Consilience Creative Program" (IITP-2018-2017-0-01015) supervised by the IITP (Institute for Information & Communications Technology Promotion); and the ICT R&D program (2017-0-00678) of MSIT/IITP. This work was supported by the Ministry of Education and Science of the Russian Federation (project No. 8.3244.2017/PCh).

**Conflicts of Interest:** The authors declare no conflict of interest.

## Appendix A. Derivation of Equation (1)

We can derive geometric relations, depending on the observation parameters, from the detailed scheme depicted in Figure A1.

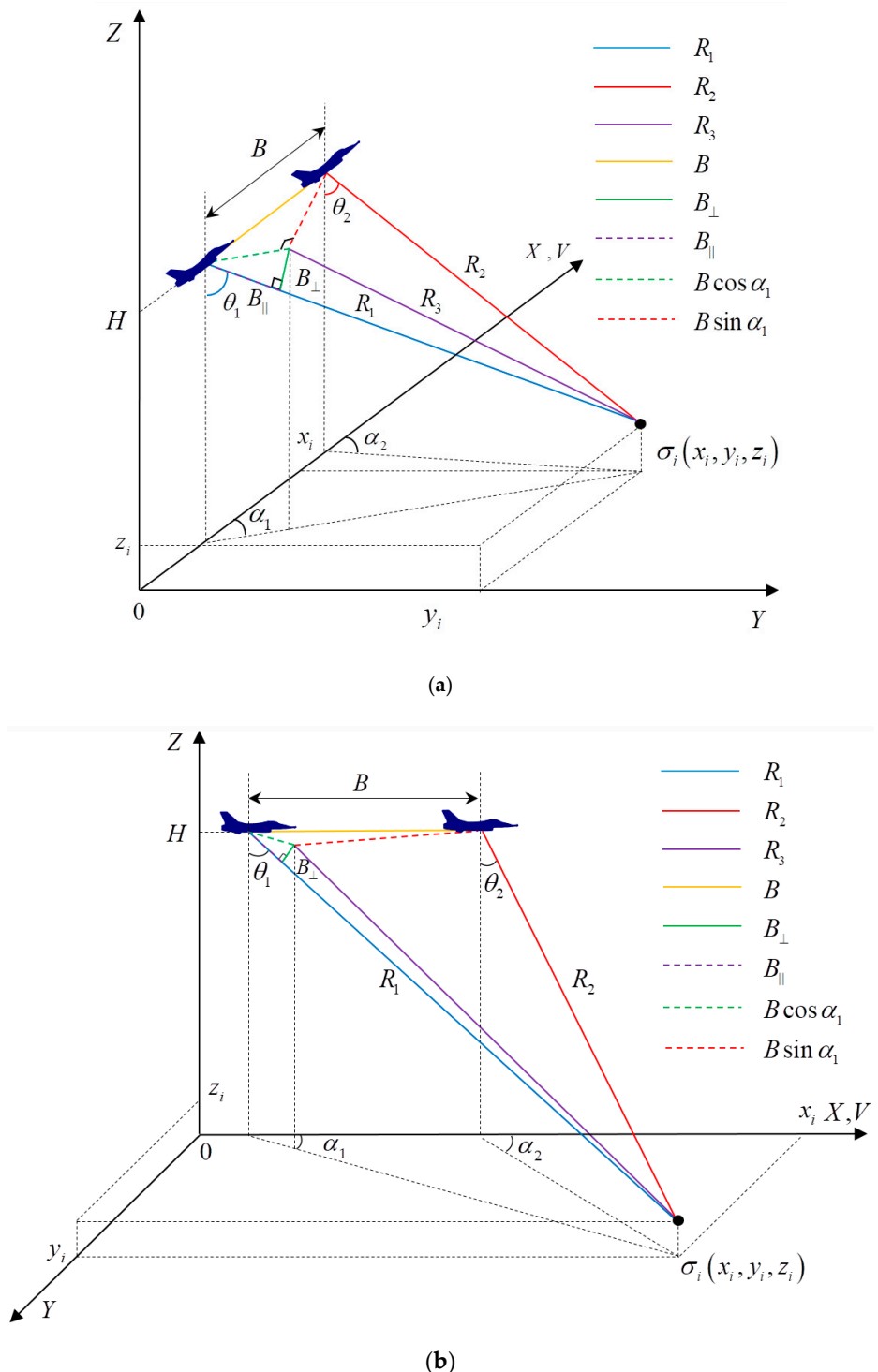

(**a**)

(**b**)

**Figure A1.** Refined observation geometry for single-antenna single-pass interferometry: (**a**) 3-D from Z-Y view; (**b**) 3-D from Z-X view.

According to Figure A1, the expression for the height of the resolution element above the reference level is:

$$z = H - R_1 \cos \theta_1 . \tag{A1}$$

Denoting $\Delta = R_1 - R_2$ as the propagation path difference of the electromagnetic waves, we can obtain the expression for the interferometric phase difference of the receiving signals at the spatially separated antennas:

$$\phi = \frac{4\pi}{\lambda}(R_1 - R_2) = \frac{4\pi}{\lambda}\Delta, \tag{A2}$$

where $\phi$ is the interferometric phase difference.

Important parameters for interferometric sensing are the perpendicular component $B_\perp$, and the parallel component $B_\parallel$, of the interferometric baseline $B$. From Figure A1 it follows that these components are defined as:

$$\sin \theta_1 = \frac{B_\parallel}{B \cos \alpha_1} \Leftrightarrow B_\parallel = B \cos \alpha_1 \sin \theta_1 \tag{A3}$$

$$\cos \theta_1 = \frac{B_\perp}{B \cos \alpha_1} \Leftrightarrow B_\perp = B \cos \alpha_1 \cos \theta_1. \tag{A4}$$

Additionally, from Figure A1:

$$\begin{cases} R_3^2 = R_1^2 + (B \cos \alpha_1)^2 - 2R_1 B \cos \alpha_1 \sin \theta_1 \\ R_3^2 = R_2^2 - (B \sin \alpha_1)^2 \end{cases} \Leftrightarrow R_2^2 = R_1^2 + B^2 - 2R_1 B \cos \alpha_1 \sin \theta_1, \tag{A5}$$

and:

$$\sin \theta_1 = \frac{R_1^2 + B^2 - (R_1 - \Delta)^2}{2R_1 B \cos \alpha_1}. \tag{A6}$$

Thus:

$$\cos \theta_1 = \sqrt{1 - \sin \theta_1{}^2} = \sqrt{1 - \left(\frac{R_1^2 + B^2 - (R_1 - \Delta)^2}{2R_1 B \cos \alpha_1}\right)^2}, \tag{A7}$$

where $\Delta = \frac{\lambda}{4\pi}$.

As a result, we can obtain the relationship between the squint angle $\alpha_1$, the look angle $\theta_1$, and the phase difference $\phi$:

$$z = H - R_1 \sqrt{1 - \left(\frac{R_1^2 + B^2 - \left(R_1 - \frac{\lambda}{4\pi}\phi\right)^2}{2R_1 B \cos \alpha_1}\right)^2}. \tag{A8}$$

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
