# Peer review of "A New Single-Pass SAR Interferometry Technique with a Single-Antenna for Terrain Height Measurements"

_remotesensing, doi:10.3390/rs11091070_

Round 1

Reviewer 1 Report

This study proposes a single-pass interferometric synthetic aperture radar imaging technique using a single antenna for estimating a digital elevation model.

The presented technique is innovative and the topic of the paper fits the scope of the journal. I have some comments the author should address before submission.

The author must  improve the technique validation section. There is no statistical accuracy assessment. I would recommend to add a scatter plot where on the X axes you have DEM values for the 2 pass interferometry while on the Y axes there are the single pass DEM values.

This would be fundamental in supporting the claim the author make about the 1.5 limiting accuracy factor.

From Figure 1 and 2 it seems the acquisition mode the authors use is spotlight i.e. the beam gets steered in order to illuminate the target for a longer amount of time when compared to the stripmap approach.

This is confusing since the author claims they used stripmap data characterized by a certain squint.

The authors should also explain how the data get focused in the two different figures.

Considering an acquisition time of 10 sec In Figure 2 you could focus the first 5 sec of data for one image and the last 5 sec. How this translates in figure 1 ?

Also, what is the accuracy difference between the two different configurations shown in figure 1 and figure 2?

Given the fact that the data the authors use in this paper are not freely available  the paper would gain value if the author added a table listing all the actual sar sensors and what would be the achiavable height accuracy using this technique.

Few more detailed comments below

Line 72 – No mention about Kondor-E satellite in reference 18 please add correct reference

Line 109-116 What is the difference between this acquisition mode and the Spotlight acquisition mode ? i.e.

Munson, D. C., O'brien, J. D., & Jenkins, W. K. (1983). A tomographic formulation of spotlight-mode synthetic aperture radar. Proceedings of the IEEE, 71(8), 917-925.

Please add differences/characteristics in common to the spotlight acquisition mode.

Line 114 – “L2 started”

Line 117 look angle were very close (of the order of how many Degrees? ) Please add.

Figure A1 B parallel is not visible in  the figure not indicated in letters

Author Response

We would like to thank the Reviewer and the Editor for reading this manuscript and for providing helpful comments and constructive suggestions. We have revised the manuscript to improve its quality and readability in accordance with the suggestions.

Our point-to-point responses to the Reviewer’s and Editor’s comments (each comment is followed by an answer) are presented below. 

Reviewer 2 Report

The authors conducted interesting and significant work. I have several minor comments:

Line 186 “where N is the number of incoherent integration” needs clarification.

Figure 4a and 5a are not very clear, why they have three lines for each interferometer

Optical image of modeled surface (Figure 6b) shows white color of the surface. Was it snow covered?

Section 8 must include specifications of SAR sensor and antenna used in experiments.

Author Response

We would like to thank the Reviewer and the Editor for reading this manuscript and for providing helpful comments and constructive suggestions. We have revised the manuscript to improve its quality and readability in accordance with the suggestions.

Our point-to-point responses to the Reviewer’s and Editors’s comments (each comment is followed by an answer) are presented below. 

This manuscript is a resubmission of an earlier submission. The following is a list of the peer review reports and author responses from that submission.

Round 1

Reviewer 1 Report

The manuscript remotesensing-447599 presents a comparison of the performance of two different SAR techniques used to retrieve the Digital Elevation Model of a portion of the Earth’ surface: 1) squint angle single-pass SAR Interferometry 2) Conventional repeat-pass SAR Interferometry.

I found the paper particularly cluttered and scattered in the description of the methodology and the presence of several language mistakes coupled with a suboptimal use of the English language makes the manuscript particularly arduous to understand. Additionally, I find the scientific content of the manuscript extremely poor. To my understanding, the main contribution of the paper is to compare the two interferometric techniques using simulations and show some squint angle InSAR data which have been previously published. The performance of both techniques has been separately described in many papers on the topic. What would have made the paper interesting would have been an analysis of the performance when the assumption of linear trajectory is violated. This is the most common case of airborne SAR Interferometry where errors and artifacts introduced by the nonlinear trajectory of the aircraft have been not described yet for the case of squint angle single-pass interferometry, at least to my knowledge.

I do not see any major scientific advancement in this manuscript.

Therefore, given the language issue and the poor scientific soundness I recommend that this manuscript is rejected.

Reviewer 2 Report

SAR images and related phase measuraments can be used in order to estmate topo and create a DEM. In single -pass interferometry generally a set of two SAR images acquired simoultaneously (no phase component due to ground deformation, no atmospheric problems etc).

A general comment

Abstract should be rewritten and include conclusions.

More info in figure's captions.

Specific suggestions/comments

 1. Introductioon line 85 "soft baseline" ?

 3. Two-passes interferometry line 138 it is not clear the shift . Is lt concern the azimuth or range?

 5. Estimation of correlation coeficients

The spatial decorrelation of course ia s parameter crucial and incident angle is directly related to the baseline. How do you take in consederation this last?

Instead of using a synthetic model of topo why non use a free DEM even with low resolution.

page 11 line 329 "PRI=60 μs" Justify this assumsion

Fig. 8 more explanation for (b) the interferogram shows a linear phase component or modulated by local topography?